# Upscaling Local Adaptive Heritage Practices to Internationally Designated Heritage Sites

Sarah Kerr [1],* and Felix Riede [1,2]

1   Department of Archaeology and Heritage Studies, Aarhus University, Moesgård Allé 20,
    8270 Højbjerg, Denmark; f.riede@cas.au.dk
2   Centre for Environmental Humanities, School of Culture and Society, Aarhus University, Moesgård Allé 20,
    8270 Højbjerg, Denmark
*   Correspondence: sarah.kerr@cas.au.dk

**Abstract:** World Heritage Sites can face an onslaught of risks from high tourist numbers, climate changes, the impacts of conflict and war, and static management practices. These sites have been ascribed a value that is considered both outstanding and universal (OUV) and as such they are placed at a higher prioritisation than all other heritage sites. The aim of this listing is to ensure their protection for future generations. Yet, the management practices enacted under this preservation mandate can be reactive rather than proactive and reflective, overly concerned with maintaining the status quo, and restricted by a complexity of national and international regulations and stakeholders. We here introduce a local-scale, community-driven heritage project, called CHICC, that offers, we argue, a useful insight into management practices that may be upscaled to internationally designated sites. Although this is not a blueprint to fit all heritage needs, some of the fundamental intentions embedded within CHICC can and perhaps should be adopted in the approaches to internationally designated site management. These include inclusivity with the local community as a priority stakeholder, a deeper understanding of the site including its future risks, consideration of the wider heritage landscape, and greater incorporation of heritage dynamism. Through analysing and evaluating the case study project, this conceptual chapter argues that adaptive heritage practices are underway in some local-scale contexts, and this can be a useful template for advancing the management of World Heritage Sites.

**Keywords:** world heritage; local heritage; climate change; adaptive management; community archaeology

## 1. Introduction

The management of World Heritage Sites is a hotly debated topic, whether it be the best approach and associated limitations, the benefits of either preservation or conservation, issues of authenticity, priorities of tourists and local communities, or lack of funds [1–5]. One particular problem which has emerged to the forefront of management debates is the limiting nature of rigid management and the need for greater dynamism within approaches and practices [3]. The call for change derives largely from greater recognition that heritage sites are not static, and so their management must be able to adapt to changes that are inevitable [6–10]. Modifications to heritage are always occurring, from biological weathering of a building's fabric to natural erosion of coastlines. There are a number of accelerants, however, which place an urgency upon the implementation of adaptive heritage practices including deterioration from high tourist numbers, destruction from conflict and war, and, as will be the focus here, climate change [6,9,11].

The case for shifting management practices from the status quo to more adaptive processes may seem like a valid and laudable course of action. Yet, putting this idea into practice is far from straightforward. World Heritage Site management has its very roots in preserving what is extant, as it exists currently, for future generations. Remaining static

is therefore implicitly called for in the management of sites. How may the practices shift robustly away from what they were intended to do? In this article, we look elsewhere for solutions, away from World Heritage Sites to those that are locally or nationally designated. Local-scale, community-driven heritage practices offer, we argue here, useful insights into adaptive heritage management. Although the management of a locally designated heritage site cannot simply be upscaled and implemented at any globally designated site, there are threads of commonality between local heritage management strategies and the called-for adaptive management of World Heritage Sites. This article will therefore analyse CHICC: a recent community-based heritage project in Western Europe. CHICC focuses on nationally designated sites at risk from climate change threats, specifically in Denmark, Ireland, and Scotland. It utilises a citizen science methodology to explore the dynamics among heritage, climate communication, and community response and—we argue—is adaptive heritage in action. CHICC's strengths in adaptive practice will be evaluated in the context of climate-change risks. Against the background of this analysis, we postulate how these may be scaled up to World Heritage Sites. We argue that dynamic and adaptive heritage practices are already taking place at a local heritage level; thus, World Heritage Site management need not reinvent the wheel but 'go small'. The three main strands within local heritage practices conducive to adaptive heritage management are: inclusive centring of community as a priority stakeholder, enhanced understanding and acceptance of heritage dynamism, and consideration of the wider cultural landscape.

## 2. Background

The term heritage is used here to include those places, monuments, sites, and areas which have been designated by some authority due to a perceived value. World Heritage Sites are those designated at the highest international level by UNESCO with the perceived value beneficial to the global community, including communities of the future. There are 1154 such international designated sites described as either cultural, natural, or mixed heritage [12]. National heritage sites, such as those selected for CHICC, have been registered by a government agency with each country creating its own list and naming conventions that are usually accessible to the public. Their perceived value rarely mentions the global community but rather is focused on national or local importance. The sites may or may not be open to the public and the overwhelming majority are not internationally designated. The categorisations are often more varying than those of UNESCO; they may specify whether the designation relates to architectural, engineering, maritime, and/or intangible aspects of heritage. Similarly, the designation does not necessarily denote protection in perpetuity; as with UNESCO, there may be different categorisations of designations with some allowing, for example, redevelopment for certain purposes.

These multiple designation systems invariably create a heritage hierarchy with World Heritage Sites prioritised over all other existing sites. This prioritisation is partially based on the ascription of value; World Heritage Site value is considered by UNESCO, and usually supported by the national listing, as both outstanding and universal (OUV). The relative nature of value thus places a comparatively diminutive valuation to other non-World Heritage Sites. Factors which support the ascription of a less-than outstanding value to non-World Heritage Sites may include small physical size of the site, little awareness of its history, or common occurrence of similar sites. This article will focus on one such smaller, 'under-valued', or 'mundane' heritage site [13,14] (p. 35) in Ireland. Doonanore Castle is a tower house (Sites and Monuments Record: CO153-015002) on the Atlantic coastline designated as a scheduled monument which identifies it as a site of pre-1700 historic importance. Such designation, however, does not offer any conservation plans, preservation order, nor funds. As a tower-house type of castle, Doonanore is one of approximately 1300 temporally and architecturally similar sites extant across the island of Ireland.

Varying designation practices identify a persistent and unresolved limitation: who determines the level of value and based on what considerations? In recent years, heritage



and value have become almost synonymous, with the former's use currently so prevalent within popular, policy, and academic discourse that Waterton et al., described it as verging on promiscuous [15]. Its widespread use has contributed to terminological vagueness and the lack of an overarching definition of its value, or rather a lack of a suitable definition. Dominant definitions have long lacked inclusivity with many voices now supporting Laurajane Smith's [16] argument that there is a Western and Eurocentric Authorised Heritage Discourse (AHD) [16–18]. This, in short, is the privileging of experts' consideration of value over those deriving from communities and other sub-national interests [19]. This singular understanding of heritage value has become ubiquitous and 'comfortable and commonplace' [20] (p. 1) and influences—if not directly establishes—governmental policy and legislation, national and internal professional agencies, and international charters. What is considered valuable has impacted directly the management of heritage, particularly what is preserved and conserved; as De la Torre [21] (p. 3) stated, 'no society makes an effort to conserve what it does not value'. As such, heritage management has become focused on retaining that which is deemed of value to a relatively small, often homogenous group of cosmopolitan 'world citizens'. The privileging of a singular AHD is a limitation which adaptive heritage management could attempt to overcome.

The overly western conception of value embedded in the AHD then became inscribed in the Venice Charter [22]. Its prioritisation of age, authenticity, and tangibility as the principal indicators both derived from and supported the tacit understanding of value amongst heritage specialists. In contrast, the public were considered little more than heritage's 'silent guest' [23] (p. 2). The origins of considering multi-vocal conceptions of value, including in their intangible forms, are often attributed to the Burra Charter, particularly its second iteration in 1999 [24,25]. The Burra Charter was the product of a shift in heritage discourse in Australia during the 1960s and 1970s [26,27] which saw greater awareness of Aborigines' rights over historic sites and artefacts. This instigated a shift, albeit a slow one, away from the Eurocentric consideration of value. Yet, indigenous people's consideration of value is not yet actively and appropriately integrated into heritage classification and there is still a way to go before their contribution to management practices can be deemed as appropriate and heritage management fully decolonised [28,29]—not least specifically in relation to climate change and heritage [30].

CHICC's research areas of Denmark, Ireland, and Scotland do not have indigenous populations; therefore, the shift, although equally as slow, has been towards greater consideration of heritage's social value [31]. This is often put into practice through a citizen science approach which strives to include non-experts in research—crucially—as active researchers and not merely an audience for results. It builds upon the foundation of outreach archaeology, such as community archaeology, field schools, and site stewardship [32] yet includes greater equality between participants and organisers and/or heritage practitioners. Although it is gaining popularity in heritage studies, it is not without its limitations (see overviews in [33,34]). There is varying consensus on appropriate levels of participation; Rivera-Collazo [35] and colleagues have established citizen scientist categorises, including contributor, collaborator, or co-creator. CHICC draws upon two of these three models: contributor and collaborator (Figure 1). With citizen science's influence upon heritage management and the study of the past, a greater diversification in practice developed along with greater engagement with citizens, professionals from other fields, and various stakeholders, including engagement with their conceptions of value. This positive development in the heritage discipline brought about a complexification of the management process: heritage managers' opinions on value are now one amongst many [21] (p. 3). Adaptive heritage management could create equity or even prioritise the conceptions of value from indigenous and local communities by building on this slow yet continuous awareness of value's complexity.

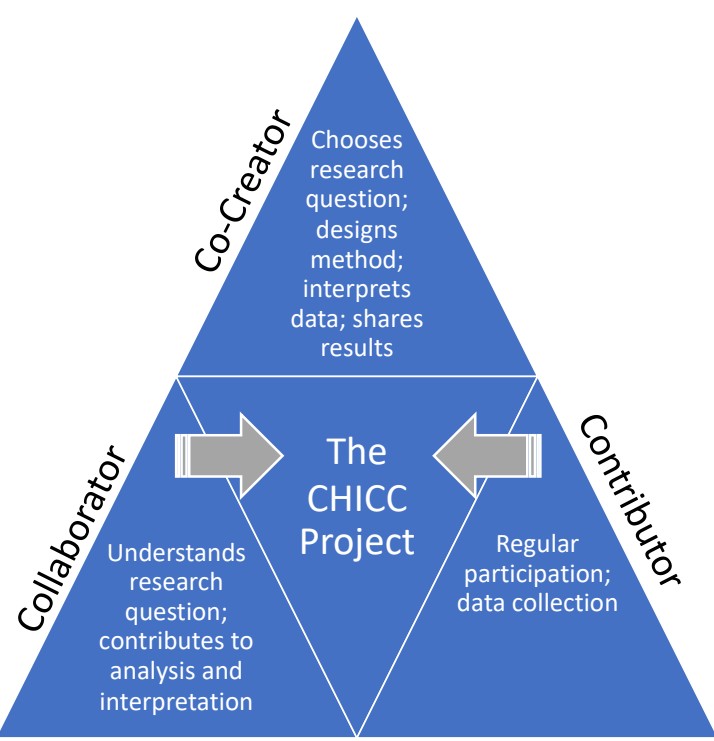

**Figure 1.** CHICC's methodology draws upon two of the three models of a community's participation in citizen science as determined by Rivera-Collazo and colleagues [35]: collaborator and contributor [Authors' own].

It is becoming commonly accepted, at least in academic discourse if not directly at heritage sites, that value is subjective; value is neither self-explanatory nor uncontested [36] (p. 466). It has been argued that value is intrinsic—and thus unchanging—whereas others believe it is bestowed and constructed from various social contexts [37,38]. Here it is considered as somewhere in between these dichotomies. Once a site, place, or area is defined as heritage, the very act of listing it as such relies on an intrinsic and tautological possession of value, even if the nature of such value is not evident [21] (p. 8). In addition, however, the constructed viewpoint equally contributes to heritage value [23]. There are 'value-formation factors' [21] (p. 8) or social processes which lead to the ascription of certain heritage values. Accordingly, some have tried to characterise heritage value; the Burra Charter, for example, lists aesthetic, scientific, and social typologies to consider whereas the Venice Charter concentrates on age and authenticity. Rather than seeing such typologies as comprehensive, these lists can be used to show the complexity of heritage and its value. Everyone connected to a heritage site, tangibly or tangentially, local or international, expert or otherwise, has slightly different conceptions of what these value types mean and how important it is to the overall value. Striving to include these various values, and accepting that there may be alternative and unknown values, is a challenge facing adaptive heritage. The first step of introducing multi-vocal values requires innovative governance, as argued by Perry and Gordon [3], comprising indigenous or local populations as equal stakeholders with governing bodies and heritage practitioners.

Managing heritage value is clearly not a simple process yet it is further complexified when the presentation of such value changes. Changes to a heritage site may derive from human intervention, such as tourism and development, or could be prompted by climate change. UNESCO has identified climate change as an area of growing concern that has already adversely affected, and will continue to affect, World Heritage Sites and their management. It not only affects the built environment but archaeological sites, terrestrial biodiversity, and glacier-adjacent cultural and natural heritage and marine biodiversity [39], the impacts upon which may be slow onset or rapid alterations. In this

volume's introductory chapter, the authors discuss how management practices may adapt when a site does change; one option involves delisting from World Heritage Site status. In the past fifteen years, three UNESCO World Heritage Sites were delisted; each the result of some form of development of the site. Most recently, Liverpool's maritime centre (UK) was delisted after planning was proposed and granted for a new football stadium, which put at risk or changed the setting of historic mercantile buildings [40]. Arguably, the removal of the listing places the remainder of the historic centre at even greater risk of development [41] (p. 37) with the downstream ramification that the management practices of World Heritage Sites have failed in their very singular aim of maintaining the site for future generations.

A further option posited as a response to changing heritage was expanding the site's designation beyond its 'patch' of identified value to include its broader landscape of influence [3] (p. 3). This prompts consideration of cultural landscapes, a term which is both a conservation category and an academic concept (see terminological issues in [42]), connecting notions of cultural heritage with biodiversity and geo-heritage (e.g., [43]). Rössler described cultural landscapes as 'the interface between nature and culture, between tangible and intangible heritage, between biological and cultural diversity' [44] (p. 334). Therefore, it stresses the importance of connections within and beyond a site's immediate boundary, whether that boundary is physical, temporal, descriptive, or disciplinary. This allows consideration of the depth of value associated with sites over time, including from indigenous and local populations and other non-expert stakeholders. This concept is strengthened by Chakrabarty's observations that research must overcome the boundary between natural and cultural histories, particularly within the context of climate change, which is a key driver of change for many heritage sites [45]. This correlates with a fundamental requirement for adaptive heritage that sites—built, natural, or cultural—must be viewed as part of their wider landscape [3] (p. 1). Embedding cultural landscapes in heritage practices may provide the opportunity to consider heritage beyond the physical landscape, to include the overlaps of associated tangible and intangible heritages, and to understand shifting values over time. This could provide a foundation for the multi-vocality and inclusivity that adaptive heritage demands.

To understand and communicate the complexity of heritage value in cultural landscapes, Actor-Network Theory (ANT) is a useful process of inquiry and descriptive method. ANT assists in understanding the associations between discrete things—or actors—and therefore is highly relevant to heritage sites, including those contested, controversial, or changing [46–50]. ANT considers that actors are connected to one another and to networks with resulting 'actor-network' connections often described as a web [51]. Crucially, for the adaptive-heritage context, an ANT approach does not consider associations as occurring only between materially homogenous actors. Rather, heterogenous elements are associated with one another, with no limit on what they may be, as Dolwick stated 'anything and everything' [46] (p. 39). This is not to say that all layered and complex attributes of heritage value can easily be captured, rather that there is a wider set of categories that should be considered, for example, beyond the indictors of material aesthetics or historic fabric. In addition to considering heritage as a multi-material web, ANT considers each element as an active node which creates and recreates the web. As such, this conceptualisation may be useful for adaptive heritage, particularly Perry and Gordon's argument for living heritage [3] (p. 7). Their argument that heritage dynamism is not only inevitable but a part of the site's value requires consideration of various actors within evolving networks.

As the impacts of climate change upon heritage became more evident, preservation and methods to do so emerged to dominate the literature. There are identifiable foci, including the quantification and mapping of sites deemed under threat [52], evaluating risk [53], sustainable preservation methods, and the subsequent impacts when preservation is not achieved, whether they are physical [54], economic, or touristic impacts [55]. Preservation narratives are strongly enshrined in governmental responses to climate threats, including in CHICC's project areas [56–58]. Alongside this, there is increasing discourse within

academic-led research which explores loss; this includes the work by Harvey and Perry [10] on managed loss and by Caitlin DeSilvey [59] on curated decay. These themes coalesce in the recent calls for adaptive heritage management, such as those led by DeSilvey [6] and Harrison [8]. DeSilvey and colleagues have called for 'adaptive release' practices within heritage management which would allow for dynamic transformation of heritage sites, including transformation of its values, within the context of its wider landscape [6] (p. 241). Their argument, which grew from the Landscape Futures and the Challenges of Change project [60], built upon earlier calls for adaptive reuse in heritage management practices. The adaptive reuse concept focused on the adaptation of heritage for contemporary requirements which arguably has now become relatively commonplace [6] (p. 420) [61].

What often remains missing across much of this work, however, is the perspectives—or voices—of the local communities who are impacted, despite the aforementioned developments deriving from statements such as the Burra Charter. To close this gap, scholars and heritage practitioners in recent years have directly approached the relationships among communities, climate change, and heritage. An example of this is the focus on the impacts of climate change on heritage and the role of communities in mitigation [62], whereas others have explored heritage and climate change through the lens of loss [10]. The discipline of heritage studies is being shaped by such discourse, particularly work on heritage decay, conservation [63], social value, and authenticity [64,65]. There is, therefore, developing energy at the nexus of climate change, heritage, and communities. This article will now turn to the Horizon 2020-funded project Culture, Heritage and Identities: Impacts of Climate Change in NW Europe (CHICC: Grant No: 895147), which focuses on this intersection. Against the canvas of one of CHICC's signature case studies, we argue that a form of adaptive heritage is in fact practised at this small scale, local-heritage-centred community archaeology project, and we identify key elements of its practises that can be scaled up to aid the management of World Heritage Sites. Although our study is neither comprehensive, canonical, or normative, we do suggest that there is great value in cataloguing and dissecting on-going small-scale heritage projects—often inclusive, innovative, and intimate—with an eye towards adaptive upscaling.

## 3. Methods

CHICC works with local communities in Denmark, Ireland, and Scotland to explore, through community archaeology and citizen science, whether heritage impacts climate communication and action. It focuses on heritage sites at risk, including those already experiencing climate-change induced impacts. These vary from sites which have already been affected in a drastic sense, such as Nørre Vosborg in Denmark, which was moved inland from the Nissum Fjord after a series of storm surges, to those mildly impacted to date yet at high future risk, such as the Wemyss Caves in Scotland. We argue that CHICC practices many of the fundamental elements outlined in the calls for adaptive heritage. As summarised in Table 1, the features of adaptive heritage argued for by Perry & Gordon [3] appear to align specifically with CHICC's centring of community as a priority stakeholder, enhanced understanding of heritage dynamism, and consideration of the wider cultural landscape. This suggests that local-scale community archaeology projects could provide a baseline from which World Heritage Site management can develop. The CHICC project is outlined here followed by an evaluation of the elements, which may be useful for adaptive heritage practice at sites of global designation.

CHICC explores the role and influence of heritage in climate communication and action. Its project areas of Denmark, Ireland, and Scotland are currently experiencing the impacts of anthropogenic climate change, including higher average temperatures, increased storm activity, and sea-level rise. These have left discernible impacts upon various types of heritage already and will continue to be a dominant challenge for the management of sites over the next half-century and likely beyond [66]. Heritage situated along the Danish, Irish, and Scottish coastlines is considered particularly vulnerable to climate change, despite the distinctive physical attributes of each area [56,67]. Coastal heritage sites will continue

to experience effects from climate threats which are both immediate and cumulative: catastrophic events, such as collapses, will increase alongside slower-onset deterioration. CHICC's objective is to create maps of sites at risk from total loss or considerable damage due to climate change through the use of a community archaeology methodology [68,69], specifically citizen science.

**Table 1.** The features crucial for adaptive heritage, as outlined by Perry and Gordon [3], are aligned with the practices of CHICC. These cannot be upscaled to approach World Heritage Site management without careful consideration, as illustrated in the non-exhaustive list of potential challenges. Suggested strategies for overcoming such challenges are included.

| CHICC Practices | Adaptive Heritage Themes | Challenges for Upscaling | Strategy to Overcome Challenges |
|---|---|---|---|
| Centres the community as a priority stakeholder | Transparency and Accountability; Innovative Governance | The size of the community at World Heritage Sites | Identify and connect with indigenous and/or local communities; Utilise appropriate digital tools |
| Enhances understanding of heritage dynamism | Living heritage | Despair at loss of site or its value | Approach loss and change with sensitivity; Highlight uniqueness and/or special characteristics by focusing on sense of place |
| Considers the wider cultural landscape | Adaptive management; Monitoring and evaluation | Disciplinary foci and hierarchies | Proactively accept the multi-disciplinary and multi-vocal nature of heritage; Embrace relationship between tangible and intangible values |

CHICC's methodology builds upon the recent increase in climate change and heritage action within research and policy sectors, as well as the increase in loss and decay discourse primarily from academic outputs. It places the role of the community in the forefront by inviting individuals to participate as active researchers or citizen scientists. Citizen science is an approach whereby volunteers are involved in a research project, according to a pre-established level of engagement, which often includes collecting data and working towards the project aim. Rivera-Collazo and colleagues [35] have established that citizen scientists may be contributors, collaborators, or co-creators. The community members involved in CHICC were invited to collect and record data (contributor) and make decisions in relation to the data and its use (collaborator); however, they did not determine the project's aims (co-creator). Figure 1, therefore, illustrates that the contributor and collaborator models established by Rivera-Collazo and colleagues [35] both influenced CHICC's methodology.

In practical terms—and under the inauspicious stars of the COVID-19 pandemic—CHICC utilised social media to invite community members to attend an online heritage workshop focused on a local heritage site at risk. In the Irish case study, the site was Doonanore Castle, a tower house located on an eroding headland extending from Cape Clear Island (Figure 2), the latter located off Ireland's southwest coast. The castle is and will continue to be at risk from climate change. It is vulnerable to sudden collapses from storm surges and high winds due to its position at the edge of the Atlantic Ocean. Indeed, the promontory which once connected the headland to the island is partially collapsed, rendering access almost impossible.

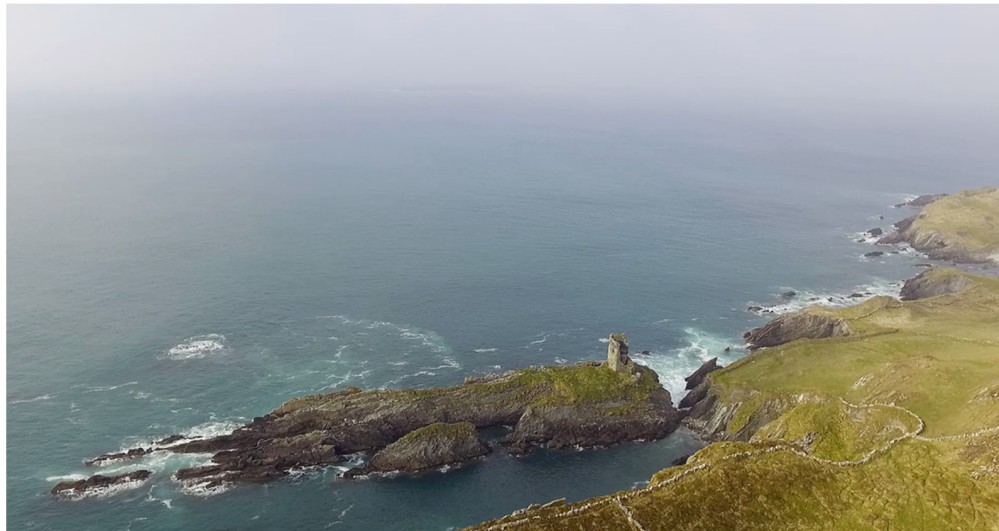

**Figure 2.** Doonanore Castle is located on an eroding headline, visible in the drone image, extending from Cape Clear Island off the south-west coast of Ireland. The castle is at risk from the high winds and waves commonly experienced on the Atlantic Ocean's coastline, particularly as they increase in intensity and frequency due to climate change (Author's own).

The workshop was also carried out online due to COVID-19 restrictions. It commenced with an informal lecture which provided an overview of the castle's development, historic climatic context, plus current and future risks from anthropogenic climate change. Both the online platform and informal nature of the talk allowed for questions to be put forth from the attendees. They could either be typed into a chat box at any point during the talk or posited in the question-and-answer session at the end; this allowed for ease of discussion between the community and the researchers, and enhanced accessibility. Attendees were provided with information about CHICC, such as its aim and duration, and what steps were required to participate. The latter included an informed consent form, contact details of CHICC researchers, and a baseline questionnaire (utilising Google Forms). The attendees were then invited to participate as citizen scientists via an email sign up with the aim of collaboratively creating a digital map detailing the past, present, and future of Doonanore Castle. The citizen scientists were invited to research and submit any information related to the castle. Drawing on ANT and conceptions of cultural landscapes, there were no restrictions on the type of information which could be included. This allowed the community to contribute their own multi-temporal and multi-disciplinary understanding of the castle. As such, data from the community were extremely diverse, including artistic depictions of the castle, personal stories of visits, photographs from the sea and mainland, poetry and lyrics, and architectural drawings. This information was input into the map alongside information deriving from the archaeologists' field-based and desk-based research on climate change and the castle's development.

The workshop was supported with surveys, also conducted online, commencing with a baseline questionnaire. The aim was to gauge the shifts in perceptions of the climate–heritage relationship and assess whether better understandings of this could facilitate action. To do so, a follow-up questionnaire was circulated five months after the first with some repeated questions. The questionnaires comprised mainly polytomous questions with a rating scale to measure the strength of attitudes in relation to the castle, its at-risk status, and climate change. This allowed the information provided to be easily converted into quantitative data which can be analysed across the submissions. To balance the lack of detail embedded within scalar questions, there were also open questions to allow for the inclusion of free text. These allowed the community to contribute more complex information and their own individual perceptions of the topic. In addition, open questions allowed a sense of greater participation for community members; this was evident in the

rich qualitative data within the answers. These provided an enhanced understanding of why the community thought their attitudes towards climate change and engagement with action have changed or otherwise.

## 4. Evaluation and Discussion

In evaluating the Irish case study of the CHICC project, we argue that it is an example of adaptive heritage practice. There are three main benefits of the local-scale community archaeology approach adopted by CHICC, which appear to adhere to the desired adaptive heritage principles and as such may be applicable to the management of World Heritage Sites. These are: centring the community as a priority stakeholder, creating understanding of heritage dynamism, and considering the wider cultural landscape. There are, however, a number of challenges when considering upscaling these practices to World Heritage Site management. These issues are discussed in the following section and summarised in Table 1 along with some possible solutions.

The community archaeology, more specifically citizen science, approach used in CHICC forefronts the community as the main stakeholder. This was established initially by selecting a heritage site likely to attract members of the local community. Doonanore is one of approximately twelve very similar tower-house castles in the area (of approximately 150 km$^2$) surrounding the main village, yet it is the only example which is both mainly extant and inaccessible. It was assumed that learning more about the castle would draw to the project those who were aware of but unable to visit it. Doonanore was built in the 15th century by the clan O' Driscoll, which remains one of the most commonplace surnames in the region; this too was considered a likely appeal to enhance the community's involvement. Thirdly, although it was not planned, COVID-19 restrictions may have generated further interest in the project. In Ireland, there was nation-wide encouragement for engaging with heritage, specifically in your local area, during 2020 and 2021 to improve mental health and reduce lockdown monotony [70]. The benefits of heritage engagement are not reliant on a pandemic and indeed there is greater awareness of the mental health benefits of engaging with heritage, particularly natural heritage sites [71–74], which is also supported in the literature (overview in [75]). These elements assisted in drawing the community to the online workshop. Yet, the success of the project lay in the subsequent prioritisation of the participants as the main stakeholders. By its very nature, citizen science is outward facing and it promotes engagement and education with those beyond the expert category [76,77]. The contributor–collaborator citizen science approach allowed the community to make genuine contributions to the project, shaping the results and steering discussions (cf. Figure 1); they were learning more about the castle while knowledge making. The benefits included creating greater awareness of the local heritage, strengthening the community's sense of place [78], and generating greater understanding of its linkages to the climate, particularly today's anthropogenic climate change.

There is a strong convergence of what are considered the key elements of citizen science and the primary aims of adaptive heritage: transparency and accountability. For adaptive heritage to actively overcome the singularity in heritage management, there must be greater communication with other stakeholders, particularly those beyond the expert categorisation. This means not only engaging with international peers but the local or indigenous communities related to the heritage site. Perry and Gordon [3] (p. 6) imply that it is not only communication which is important but clear communication, as this may equip the community with the knowledge required for engaging with the management process. Through practicing equal partnership, CHICC's dissemination channels were created from the outset and were often continuous and informal. Citizen science provides the toolkit for this communication to be effective as the participants are generating knowledge themselves to share with others, creating an enthusiasm to continuously engage with communication channels, such as CHICC's maps.

Although there is no action planned to 'save' Doonanore from climate change, citizen science allows the community to be part of the conversation on its future. This was the first

instance that the community has been told, from a so-called expert, that Doonanore was at an increasing risk from climate change. This taking place during the workshop and map creation allowed for clear, accurate communication. If discussion of protecting Doonanore were to occur in the future, the foundation for inclusive and transparent communication has been created. A limitation that could impact this is the poor maintenance of communication once the project ends. This is an issue detected in many citizen science projects [77] (p. 613).

How this could be upscaled is problematic primarily due to the vastness of the World Heritage Site's stakeholder group: who constitutes the World Heritage Site's community? In UNESCO designation, it is evident that the entire world, including future generations, are deemed the community, which is clearly not practical for an in-person citizen science approach. The digitality of CHICC may provide possible solutions. Although this was obligatory due to COVID-19 restrictions, the online workshop, lecture, questionnaires, and map creation arguably enhanced the project's inclusivity and accessibility from beyond the immediate locality. Given the perceived global nature of a World Heritage Site's community, digital tools may provide a useful concourse to overcoming practical limitations. Moreover, recent heritage discourse has made clear that all heritage sites have a related local community, whether indigenous or non-indigenous [75,79]. What connects these groups is that they are not solely, if at all, composed of heritage practitioners or other types of so-called experts. The Doonanore community brought to the project their own perspectives of the site and its value and meaning to them, their families, and neighbours, free from bureaucracy or disciplinary boundaries. Introducing such perspectives allows for a multi-vocal understanding of heritage and its values, whereas equal partnership between the community and heritage practitioners could create the platform for transparency and accountability. Community archaeology, specifically with a citizen science ethos of equal partnership and inclusion, appears to be adaptive heritage in practice.

A second aspect of the local-scale community archaeology project that contributes to adaptive heritage is the ability to enhance understanding of heritage dynamism. The ways in which Doonanore Castle has changed over the course of its existence were evident in the information gathered by the citizen scientists and the archaeologists. For example, the bombardment of the castle walls from canon fire in the early 16th century created damage to its east façade, which remains visible from Cape Clear Island today. Although this was not related to the current drivers of change, it demonstrated that the castle was not always a ruin and as such may not remain as a ruin. The impacts of climate change upon the site were also explicitly evident through the information gathered. The headland upon which the castle sits was once accessible from the island via a connecting neck, yet the neck collapsed, rendering the castle only accessible to those willing to scale the rocks emerging from the Atlantic Ocean (Figure 2). Although this information was widely known throughout the community, discussions around historic photographs and engravings showing the neck in situ brought the changeability of the landscape to the fore.

The process of Doonanore's local community gathering their own evidence arguably made the information relating to heritage dynamism more explicit or at the very least more personal. In the open answer sections of the questionnaires, people included details of the climate and how it has changed in the context of the castle. One person commented on visiting the castle at low tide, describing an unusually calm winter's day, noting that only for this did they feel confident enough to scale the promontory to the castle. Such instances of the weather being embedded in the community-gathered information was common, particularly within art and poetry. In contrast to the previous account, these were usually stormy descriptions or renderings of the at-times violent Atlantic coastline. These contrasting inclusions in the map highlighted the driver of Doonanore's dynamism: the weather. It demonstrated that the Atlantic coast weather was always a threat to the extant fabric of the castle, even before anthropogenic climate change became an accelerator.

This understanding of the site's dynamism may provide the first step towards accepting it as a future inevitability. Communicating to the community, through the use of national climate data and global climate-change warnings [56,80], that climate change will

certainly affect the castle could have been received with antagonism or despair. However, by situating this information as part of Doonanore's story or life biography, it may have seemed less appalling. This was likely supported by the fact that there is a strong fishing economy in the community, plus a network of occupied islands, and so the impacts of climate change particularly in relation to storms were not entirely new information. For this approach to be scaled up to World Heritage Sites, greater caution in discussing climate change may be required, as it can lead to fatigue or fear and thus disengagement [81].

Accepting such dynamism will be central to adaptive heritage, in terms of the changing of both the heritage site and the management practices. The creation of the map with the community appears to suggest that Doonanore's dynamism is accepted locally as part of its value. Certainly, its historic dynamism, such as the canon fire and the collapse of the promontory, ignited ongoing interest in the site, with the former included in its national designation. Whether future dynamism will also be considered part of its value cannot be accurately stated. There was, however, a perceptible sense of place deriving from the community and the information it brought forth. Sense of place has been described as a vague, or even useless, term (see overview in [82,83]); yet it is important in heritage discourse as it refers to the meanings and value ascribed upon a place by an individual or group [84]. Although it was not the focus of the CHICC project, it became apparent through joining the community, as they researched and discussed their local heritage, that they deemed Doonanore special. Special, following Schofield and Szymanski's definition, does not necessarily mean iconic or globally significant but a feature deemed of value to the local community and that contributes to setting the area apart from others [82] (p. 2). The community members appear both proud of and affectionate towards the castle. This is evident in the enthusiasm for joining the project and the wealth of information brought forth. It is possible that part of what makes Doonanore special is its current and future dynamism; its at-risk status sets the castle apart from the numerous similar buildings in the surrounding area, and the genuine danger of visiting the site makes it a point of local interest as it is often viewed only from the safety of boats. This implies that the castle will remain special to the community regardless of how the castle will change now and in the future.

Including sense of place in community engagement to support adaptive heritage management of World Heritage Sites may be difficult as much sense of place research focuses on local communities and their local areas. This appears to be because some have outlined that a feeling of belonging is pertinent to the development of a sense of place relationship. At Doonanore, it was likely the localness promoted an acceptance of dynamism, something which would not be extended from the Doonanore community to other heritage sites. As stated earlier, however, each World Heritage Site does have a community, whether local or indigenous; indeed, Hilary Orange explored sense of place in relation to the UNESCO World Heritage Site of the Cornish mines (UK) [85]. As such, striving for greater understanding of heritage dynamism may be intertwined with centring the local community as the priority stakeholder.

The third element of the CHICC project which may support adaptive heritage is the consideration of the wider cultural landscape. There were no restrictions placed upon the type of information which could be brought forth into the map. Therefore, CHICC embraced from its outset that the castle's value did not end at its walls nor the extent of the headland. It did not end at the cliff of the island from which the headland extended or even at the main village on the Irish mainland. There was considerable connectivity embedded in the information brought forth; for example, the links between the castle and a contemporary castle, Dunalong, on a neighbouring island, which seemingly worked alongside Doonanore to manage the fishing and victualing in the bay. One of the poems submitted was the Sack of Baltimore by Thomas Davis (1814–1845), which refers to the area as Carbery's hundred isles, a moniker which is locally and nationally recognised despite there not being one hundred islands in that area. This brought into the cultural landscape of Doonanore not only the physical landscape of the area with its islands and coastlines but

the history and literature of the wider county of Cork. The information submitted exceeded the expectations of how large the castle's cultural landscape is. Submissions included notes and photographs from previous visitors to the region who had viewed the castle from the ocean during tours or fishing trips. These derived from North America, South Africa, and New Zealand while artistic depictions of the castle were submitted from Germany.

Embracing cultural landscapes in CHICC highlights two aspects which may be useful for adaptive heritage management. Firstly, this approach actively recognises the multidisciplinary nature of heritage and embraces the overlaps between tangible and intangible heritages. Although a tangible castle was the focus, its value to the community is perceptible and enjoyed through other heritages, such as the other castles or literature. Furthermore, it has shown that mundane local heritage is not only special and of value to the local community but those non-local visitors who interact with it briefly and possibly only once. This suggests that heritage management practices with cultural landscapes embedded could be scaled up to approach World Heritage Sites.

## 5. Conclusions

The challenges facing World Heritage Sites have accelerated discussions on heritage's dynamic nature and generated an impetus to address restrictive heritage management practices. In this issue's introductory chapter, Perry and Gordon [3] call for a proactive shift away from the limiting and static practices, establishing that heritage management must be adaptive and explicitly recognise the changeability of heritage and its values. They set out the central themes to an adaptive management approach, outlining the importance of each. In this article, we looked towards locally and nationally designated heritage sites and their management practices, arguing that one local-scale community archaeology project, with a citizen science approach, may provide a blueprint for achieving such themes. CHICC's inclusive centring of community as a priority stakeholder, creation of heritage dynamism understanding, and consideration of the wider cultural landscape all correlate with a more reflective and inclusive conception of heritage and could contribute to enhanced—adaptable and adaptive—management practices.

Upscaling such ideas to World Heritage Sites would not necessarily be straightforward; as such, we put forth this case study provisionally not prescriptively. The citizen science approach of CHICC required suspending all notions of AHD and attendant ideas of value. Instead, what the community considered of value and special was included and viewed as equally important to the castle's tangible remains. Although the heritage sector has made strides in moving away from AHD, the normative mould of World Heritage status and value ascription creates a notable barrier to further progress. Therefore, to apply these practices at a globally designated site, the World Heritage designation and management process may need softening or even—controversially perhaps—a transferral from the expert to the community. These issues of upscaling these practices are neither trivial nor miniscule; they are not, however, insurmountable. This conceptual chapter has argued that adaptive heritage practices are underway in the local-scale, citizen science-driven, and mixed analog/digital CHICC project, and likely many other projects of a similar ilk. These can be useful starting points, or even templates, for the adaptive management of World Heritage Sites.

**Author Contributions:** Conceptualization, S.K. and F.R.; methodology, S.K.; investigation, S.K.; resources, S.K.; data curation, S.K.; writing—original draft preparation, S.K. and F.R.; writing—review and editing, S.K. and F.R.; visualization, S.K. and F.R.; supervision, S.K. and F.R.; project administration, S.K.; funding acquisition, S.K. and F.R. All authors have read and agreed to the published version of the manuscript.

**Funding:** This research was funded by Marie Skłodowska-Curie Actions (MSCA) under the European Union's Horizon 2020 research and innovation programme (grant agreement 895147).

**Institutional Review Board Statement:** The project is approved in accordance with Aarhus University's guidelines for the university's Research Ethics Committee (IRB).

**Informed Consent Statement:** Informed consent was obtained from all subjects involved in the study.

**Conflicts of Interest:** The authors declare no conflict of interest.

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
