# Peer review of "Upscaling Local Adaptive Heritage Practices to Internationally Designated Heritage Sites"

_climate, doi:10.3390/cli10070102_

Round 1
Reviewer 1 Report
The paper is based on a thorough methodology and extensive references. The strong theoretical framework makes the study somewhat abstract, which could be made more plastic by better positioning the case study.
The study examines a specific case study in a strong theoretical framework. In the introduction, it would be important to formulate the research question more clearly and emphatically and then to introduce the case study as well. All this can also affect the Chapters Background and Methodology, the question of the research and the examined example should be positioned more clearly in front of the theoretical background.
It would be important to improve the illustration of the text. The small number of figures is not enough to illustrate the case study and the topic. When editing the table, some cells are missing page lines and need to be fixed.
Author Response
Dear Reviewer,
Many thanks for your considered comments on the paper. The following changes have been made as per your recommendations.
- We have introduced the case study and aim more clearly in the beginning of the paper. See lines 60-63. The project has also been mentioned in the abstract, as mentioned in another comment, see lines 18-22.
- Issues relating to the visual aspects of the table have been rectified (lines missing, spaces too large). See page 8.
- The sentence structure which mentioned war impacting heritage has been changed to shift the emphasis to the topic under consideration. See lines 11-12.
- The ambiguity in relation to the influence of the co-creator citizen science model (mentioned at numerous points and within figure 1) has been clarified. See lines 151-154, 165-166 and 328-333.
- More relevant detail on the workshop has been added throughout the methods section. See pages 9-10.
- A number of highlighted errors have been corrected – for example, date missing on Perry and Gordon reference.
With best wishes from the authors.
Reviewer 2 Report
· Line 82, paragraph 2. Should be “maritime and/or intangible” and not “maritime and/ or intangible”.
· Line 102, paragraph 2. This line is not needed.
· Line 160, paragraph 2. The Figure 1 is not in the center.
· Line 160, paragraph 2. In this figure (Figure 1), what is the relationship of the co-creator with the CHICC project? It's not clear in the schematic
· Line 287, paragraph 3. It is possible to have a smaller table? It is too wide.
· Line 287, paragraph 3. Borders are missing in some of the table cells.
· Line 361 and 362, paragraph 3. These lines are not needed.
· Line 512, paragraph 4. This line is not needed.
· Line 544, paragraph 5. The year is missing in the bibliographic reference of Perry and Gordon.
· Please check the hyphenation all over the text. Example: Line 79, paragraph 2
· In the abstract you talk about the risk of war to the heritage, but during the text this case is rarely mentioned; if it's not that important, why is it in the abstract?
· In the abstract it is not explicit that the article is developed around a specific case study.
The article essentially presents a case study. However, the article is a bit "closed", as it does not make many comparisons with the results, for example, of other existing projects in the world.
However, without a doubt, it was a great read. Congratulations.
Author Response

(The authors gave the same response as above.)

Reviewer 3 Report
Dear authors,
Your manuscript provides a compelling narrative on the threats that heritage loss and damage due to climate change pose on the livelihoods of local communities and cultural identities, therefore it provides clear take-away messages for both scientists and policy-makers. I would recommend to better showcase the methodology at hand and provide further insight into the way you constructed your workshop, purposed it, engaged with citizens as scientists. I think that the readership could take better stock of your study if these details are to be included in your paper.
Author Response

(The authors gave the same response as above.)
